# Synthetic Bone Substitutes and Mechanical Devices for the Augmentation of Osteoporotic Proximal Humeral Fractures: A Systematic Review of Clinical Studies

**DOI:** 10.3390/jfb11020029

**Published:** 2020-05-05

**Authors:** Giuseppe Marongiu, Marco Verona, Gaia Cardoni, Antonio Capone

**Affiliations:** Orthopedic and Trauma Clinic, Department of Surgical Sciences, Cagliari State University, 09126 Cagliari, Italy; marco.verona@tiscali.it (M.V.); gaia.cardoni@gmail.com (G.C.); anto.capone@tiscali.it (A.C.)

**Keywords:** proximal humeral fracture, augmentation, bone substitutes, cement, polymethylmethacrylate, PMMA, calcium sulfate, calcium phosphates, hydroxyapatite

## Abstract

Background: Different augmentation techniques have been described in the literature in addition to the surgical treatment of proximal humeral fractures. The aim of this systematic review was to analyze the use of cements, bone substitutes, and other devices for the augmentation of proximal humeral fractures. Methods: A systematic review was conducted by using PubMed/MEDLINE, ISI Web of Knowledge, Cochrane Library, Scopus/EMBASE, and Google Scholar databases according the Preferred Reporting Items for Systematic Reviews and Meta-Analyses (PRISMA) guidelines over the years 1966 to 2019. The search term “humeral fracture proximal” was combined with “augmentation”; “polymethylmethacrylate, PMMA”; “cement”; “bone substitutes”; “hydroxyapatite”; “calcium phosphates”; “calcium sulfate”; “cell therapies”, and “tissue engineering” to find the literature relevant to the topic under review. Results: A total of 10 clinical studies considered eligible for the review, with a total of 308 patients, were included. Mean age at the time of injury was 68.8 years (range of 58–92). The most commonly described techniques were reinforcing the screw–bone interface with bone PMMA cement (three studies), filling the metaphyseal void with synthetic bone substitutes (five studies), and enhancing structural support with metallic devices (two studies). Conclusion: PMMA cementation could improve screw-tip fixation. Calcium phosphate and calcium sulfate injectable composites provided good biocompatibility, osteoconductivity, and lower mechanical failure rate when compared to non-augmented fractures. Mechanical devices currently have a limited role. However, the available evidence is provided mainly by level III to IV studies, and none of the proposed techniques have been sufficiently studied.

## 1. Introduction

The incidence of proximal humeral fractures (PHF) is increasing with the growth of the elderly population [1]. Among the geriatric population, upper extremity fractures account for one-third of the total incidence of fragility fractures and represent the fourth most common cause of hospitalization [2,3]. 

In the literature, no consensus exists with regard to the optimal fixation strategy for the treatment of osteoporotic proximal humeral fractures. Different techniques have been proposed in the literature, including intramedullary nails, locking plates, percutaneous pinning, tension band wiring, and arthroplasty, but the ideal approach is yet to be defined [4,5]. Locking plate fixation with multiaxial screws is considered the most suitable procedure, particularly for multi-fragmentary fractures, but according the literature, it comes with a complication rate up to 49% [6,7,8,9,10,11]. The postoperative implant loosening rate was reported in the range of 14% to 22.2%, and the reoperation rate is up to 29% [12]. The most frequent complications are screw perforation and varus collapse of the head.

Low bone mineral density (BMD) and lack of medial support have been identified as the two main causes of the failure of the treatment [13,14]. Low BMD is highly correlated to the pullout strength of the screws, and therefore it has a major effect on the stability of the fixation and the biomechanical behavior of the bone to implant interface [15]. Gardner et al. first described the correlation between a lack of medial support and the loss of reduction after fixation [16]. Jung et al. found a significantly worse result in patients with comminution of the medial hinge [17]. Clinical and biomechanical studies have focused on finding a solution to overcome these problems and to enhance plate fixation. New plate design with poly-axial screws and with medial column support screws have been studied; however, the benefits of these designs are still to be defined [18,19,20].

Plate fixation has been associated with different techniques of bone grafting augmentation, including both autograft and allografts, with the purpose of addressing the need for medial support and fill the void after osteoporotic fractures [21,22,23,24]. Despite encouraging bone healing rates and good clinical results reported in the literature, possible limitations to the extensive use of these methodologies are represented by limited accessibility to allografts and increased costs if not derived by inhouse bony banks [21]. Almost all allografts undergo some form of processing prior to their use such as deep-freezing (−70 °C), freeze drying (lyophilizing), gamma-irradiation (standard dose 25 mGy), and fluid pressurization with saline solution or sterilizing agents [25]. Nevertheless the potential risk of infection, transmission of diseases, and the immunogenicity of the grafts still remains [26,27].

Different options such as bone substitutes, cement, and metallic devices have been proposed as alternative treatments for the augmentation of osteoporotic proximal humeral fractures, in association with plating and other fixation techniques [28,29,30].

The aim of this systematic review of the literature was to analyze the clinical application of cements, bone substitutes, and metallic devices for fracture augmentation in patients affected by osteoporotic proximal humeral fractures.

## 2. Materials and Methods

To conduct a comprehensive study of the evidence, we performed a systematic review with narrative review of the literature. Our review was performed according to the Preferred Reporting Items for Systematic Reviews and Meta-Analyses (PRISMA) guidelines [31]. The protocol of the review was not registered due to its small size.

We identified a precise population, intervention, comparator, outcome (PICO) question. We studied patients with osteoporotic humeral fractures who had received surgical reduction and fixation associated with different augmentation techniques: cementation, bone substitutes, and mechanical devices. The primary outcomes were mechanical failure and complication rates compared to not-augmented fractures. Clinical results in terms of functional scores were also reported.

The databases of PubMed/MEDLINE, ISI Web of Knowledge, Cochrane Library, Scopus/EMBASE, and Google Scholar databases were comprehensively searched for a review of the literature. The search strategy was the combination of the keywords (humeral fracture proximal) AND (bone substitutes OR augmentation OR hydroxyapatite OR cement OR PMMA (polymethylmethacrylate) OR calcium sulfate OR calcium phosphate OR cell therapy OR tissue engineering). The search date was limited to 31 December 2019. All titles and then abstracts were reviewed for relevance. Those considered consistent to the stated purposes of this review were read in full text and had their information extracted. Inclusion criteria were clinical studies including patients with osteoporotic proximal humeral fractures, with minimum 12 months mean radiological and clinical follow up, reporting complication and reoperation rates. Exclusion criteria were biomechanical studies, computational and finite element analysis, and other non-clinical applications. Moreover, non-English language clinical studies, case reports, and grey literature were also excluded. At least two investigators evaluated each article (G.M. and G.C). If there was disagreement between reviewers, any discrepancies were resolved by a third reviewer (M.V.). Our search strategy is shown in Figure 1.

Quality assessment was performed using the Coleman Methodology Score (CMS). The score can be rated as excellent (85–100 points), good (70–84 points), fair (50–69 points), or poor (<50 points) [32].

## 3. Results

Out of 973 studies, 426 duplicates were excluded, 511 studies were excluded according to the screening process, and 36 full-text articles were assessed for eligibility. Subsequently, 28 studies were excluded after full-text analysis due to the exclusion criteria. Two studies were included after the analysis of the reference lists of the selected papers. Therefore 10 clinical studies, with a total of 308 patients, were included. Mean age at the time of injury was 68.8 years (range of 58–92) and 166 out of 308 (54%) were female.

The articles included in the review are listed by topic in Table 1. Clinical evidence ranged from level I to level IV (level of evidence (LoE) I: 1, LoE II: 1, LoE III: 5, LoE 3: 10) according to the Oxford Centre for Evidence-Based Medicine. According to the CMS scoring system, 10 studies revealed a poor, 6 studies a fair, and 1 study a good methodology. Due to the substantial study heterogeneity and small sample sizes, the data obtained from the selected studies were not adequate to perform a meta-analysis. For these reasons, a descriptive approach to data analysis was performed. Generally, in the studies analyzed, the main goal of fracture augmentation was to provide mechanical support to the osteoporotic bone rather than biological supply and bone remodeling. The most commonly described techniques were, in fact, *reinforcing the screw-bone interface* with bone cement, *filling the metaphyseal void* with synthetic bone substitutes, and *enhancing structural support* with metallic devices. 

### 3.1. Synthetic Bone Substitutes and Cement Augmentation

The properties of the ideal biomaterial for bone augmentation should include both mechanical properties (void filling capacity, structural support, and fixation enhancement) and biological properties (osteoconductivity, osteoinductivity, osteogenicity). However, none of the biomaterial available in clinical practice completely covers all these characteristics. Composites for augmentation are available in form of injectable materials that harden in situ or granules, chips, and blocks. In the case of proximal humeral fractures, bone substitutes used in clinical settings are mostly injectable cements, which differ from each other for mechanical and biological properties (Table 2).

#### 3.1.1. Polymethyl Methacrylate (PMMA) Cement

In the clinical setting, PMMA has been used to improve screw fixation in osteoporotic bone such as wrist and proximal femur fractures, in order to fill subchondral voids in tibial plateau fractures and provide structural support in metaphyseal fractures or vertebral fractures [4,47,48]. Three studies reported the results of polymethyl-methacrylate (PMMA) injectable cement for PHF augmentation [30,31,32,33,34], for a total of 96 patients with a mean age of 75.86 years (range of 64–92), with the purpose of reinforcing the screw-bone interface after plating fixation. In all the three studies, the same PMMA (Trauma Cem V; DePuySynthes, West Chester, PA, USA) was used. Cannulated screws were used to fix the fracture to the plate, then prefilled syringes with PMMA were used to augment cannulated screw with 0.5 to 1 mL of cement. The setting time of the PMMA is 15 min at body temperature, and after curing, the compressive strength is around 85 MPa.

In 2018, Katthagen et al. [30] prospectively treated 24 proximal humeral fractures with the PHILOS plate (DepuySynthes, West Chester, PA, USA) and additional humeral head screw augmentation with PMMA (Figure 2).

At the follow-up, no screw penetration was reported. The authors compared the results of 24 fractures derived from a historic retrospective cohort of patients, matched for gender, sex, and type of fracture, who received the conventional PHILOS plate without cementation. In the non-augmented group, four patients (16.6%) suffered screw penetration after the first 6 months, with a significantly increased risk of early loss of reduction (*p* = 0.037). In the augmented group, no implant-related complications were observed, but two patients (8%) had major biological complications (one avascular necrosis of the humeral head and one nonunion). After 12 months, the mean constant score was 72.9 ± 18.1 points in the PMMA-augmented group and 73.0 ± 13.1 points in the non-augmented group (0.557), with no significant differences observed (*p* = 0.62).

In 2019, Siebenburger et al. [33] reported the results of a retrospective comparative study, in which 55 patients underwent in order to open reduction and plate fixation for PHF. In 39 cases, the screw-tip augmentation with PMMA was performed. The mean constant score was 63.7 in the augmented group and 62.6 points in the non-augmented group The authors did not find a statistically significant difference in terms of clinical outcome. The overall complication rate was 16.3% in the locking plate-only group, compared with 12.8% in the group with screw tip-augmented osteosynthesis (*p* = 0.086). The loss of fixation occurred in 10.9% vs. 5.1% (*p* = 0.074). However, these differences were not statistically significant.

More recently, Hengg et al. [34], in a multicenter randomized controlled trial, enrolled 67 patients older than 65 years with displaced or unstable PHF. They compared the risk of mechanical failure during the first year after PHILOS plating without (control group, 34 patients) and with (augmented group, 33 patients) screw augmentation with PMMA. In total, nine eligible patients (13.4%) patients had mechanical failures within the first year after treatment (nine with loss of reduction, four with humeral head impaction, one with screw/plate loosening, and five with secondary screw perforation). The authors did not find any statistically significant differences in the occurrence of mechanical failures between the two study groups (augmented group, 16.1% vs. control group, 14.8%). The relative risk (95% CI) for the augmented group was 1.09 compared to the control group (*p* = 1000). The constant score in the augmented group was 66.6 points (58.7; 74.6), whereas in the not-augmented group was 64.4 (56.8; 71.9), and no significant difference was reported (*p* = 0.665). Moreover, no statistically significant differences in adverse events were observed between study groups at a 1 year follow-up. Due to these preliminary results, the study was prematurely terminated.

Several possible limitations to PMMA augmentation have been reported [48]. The main concern is related to the high temperatures reached by PMMA during the polymerization phase (100 °C), which could potentially cause bone and cartilage necrosis and subsequent fixation loosening [49]. Experimental models reported that intraarticular and subchondral temperature during PMMA curing reached levels ranging from 38.3 to 43.5 °C, which are lower than temperatures at risk for bone damage [50]. Other issues include the possibility of bone cement leakage and the interposition at the fracture site with consequent bone healing retardation [51,52]. However, none of these adverse events were reported in the studies included in the review. Furthermore, PMMA is bioinert, does not have osteoinductive properties, also is not integrated and reabsorbed by the bone. Therefore, this could represent an additional issue if revision surgery is needed.

#### 3.1.2. Calcium Phosphate Cements

Calcium phosphate (CaP) cements are the most used bone substitutes in trauma surgery and have been mainly applied to filling metaphyseal bone voids, particularly in tibial plateau fractures [53]. The augmentation of head screws applying the calcium phosphate cement either directly in the humeral head or via cannulated screws is currently performed in clinical practice. Compared to PMMA, calcium phosphate cements reach lower curing temperature and have lower compressive strength (ranging from 36 to 66 MPa) [54]. CaP cements can be reabsorbed and replaced by cancellous bone within 6 months to 10 years, which is why they lack predictable osteoconductive properties [55].

Two studies reported the results of injectable CaP for PHFs treated with plate fixation or cannulated screws, for a total of 52 patients and a mean age of 64.1 (range, 22–84). Robinson et al., in 2003 [35], treated 25 patients with severely impacted valgus fracture of the proximal humeral head by internal fixation and augmentation with injectable calcium phosphate (Norian Skeletal Repair System (SRS)). The composite is a moldable and biocompatible calcium phosphate that sets at body temperature into carbonated apatite. It has a compressive strength of 50 MPa, which is 4 to 10 times greater than the average 5–15 MPa of cancellous bone. The surgical technique consists first in the reduction of the fracture and then the injection of the composite. It is applied in a in a semi-liquid form inside the bone void, which hardens with a slower curing time than the PMMA cement (15 min circa). Mean 8 mL (range 5 to 10 mL) of composite was injected. The fracture was then fixed with non-locking buttress plate (11 patients) or cannulated screws (14 patients). The process was performed under fluoroscopy in order to avoid extravasation of the cement in the soft tissues. At 1 year radiologic follow-up, all reductions were maintained and no sign of osteonecrosis was reported. At 12 months, the mean constant score was 80 points. Egol et al. in 2012 [36] used the same setting with another calcium phosphate cement (Hydroset; Stryker, Mahwah, NJ, USA) in 92 patients who received a locked compression plate (PHILOS Synthes, West Chester, PA, USA) for PHFs. A total of 27 of these fractures were augmented with 10 mL of calcium phosphate cement, 29 were augmented with allograft cancellous chips, and 36 were not augmented. No mechanical failures were reported in the calcium sulfate group. In comparison, a significantly higher rate of screw penetrations were found in the allograft cancellous chips augmentation group (4 out of 29, 13.8%) and in the non-augmented group (7 out of 36, 19.4%; *p* = 0.02). Only one complication occurred in the cement-augmented group due to deep infection. The authors did not report the functional outcome of the patients using clinical scores, pain scales, or range of movement analysis.

One of the potential limitations for the use of calcium-phosphate cements is the lower compressive strength compared to PMMA. However, CaP cements have been widely used for filling a subchondral void in tibial plateau and vertebral fractures, showing good compression strength even after full weight bearing [56,57]. Considering the fact that the proximal humerus is not subject to axial compressive loading, as vertebral bodies of proximal tibia are, CaP cements could be considered enough to support the mechanical stability of a metaphyseal humeral fracture. CaP cements are gradually resorbed over time. Resorption time differs according to its manufacturing process (i.e., crystallinity, sintering temperature), porosity properties of the CaP cement, and surface area of injection, ranging from 6 to 10 months and representing a lack of osteoconductive properties. Therefore, the injection of a large volume of cement that could represent an inert obstacle to the bone healing process should be avoided, particularly if used to fill wide bone voids [58]. In case of massive bone loss, autologous or allogenic structural bone graft should be considered as a first choice treatment [59,60].

#### 3.1.3. Calcium Sulfate Cements

Calcium sulfate (CaS) has been used successfully as a bone graft for treatment of contained subchondral bone defects in the form of pellets or injectable graft for tibial plateau fractures [61]. CaS combines biodegradability, osseointegration, and osteoconductivity, but its low mechanical resistance limits the range of its applications in orthopedics [62,63]. However, CaS injectable composites have several potential advantages compared to calcium phosphate products such as faster curing time (range of 2 to 5 min), hardening without producing heat, and a compression strength more similar to cancellous bone (varying from 10 to 40 MPa). CaS cements are completely reabsorbed and replaced within 6–12 weeks, which is why they are considered to have good osteoconductive properties [55]. For this review, we considered two eligible retrospective comparative studies and one retrospective series. Here, a total of 65 patients received plate fixation for PHFs and augmentation with three different types of CaS injectable cement. The mean age among the studies was 65.47 years old (range of 63–86).

Lee and Shin in 2009 [37] retrospectively evaluated the radiological and clinical outcome in a cohort of 44 patients with 45 PHFs treated with plating fixation. Here, 14 out of 45 fractures were augmented with an injectable calcium sulfate graft (MIIG 115; Wright Medical Technology, Arlington, USA). The aim was to provide structural support to the reduction and fill the metaphyseal void. The MIIG 115 is a moldable cement, which has a low compression strength (≈15 MPa in dry conditions after 60 min) and it requires a setting time of approximatively 3 min and 1 min of injection time [64]. All fractures healed in both groups. Only one patient (7.1%) of the augmented group suffered a reduction failure compared with the four (12.9%) of the non-augmented group. The functional outcome was assessed using the University of California at Los Angeles (UCLA) shoulder score—the augmented group had a mean score of 30.2 versus the 28.9 points of the non-augmented group. However, the differences were not statistically significant (*p* > 0.05) and the augmentation was not randomized. Later, Liu et al. reported similar results in a retrospective comparative cohort of 50 patients older than 60 years with fragility PHFs [38]. Here, the MIIG X3 injectable calcium sulfate graft (Wright Medical Technology, Arlington, USA) was used for 29 patients in the augmented group. The other 21 patients received a PHILOS plate alone. Compared to MIIG115, MIIG X3 has higher compression strength (≈40 MPa in dry conditions after 60 min), higher viscosity, longer setting time (≈11 min), and injection time (≈2–3 min). This allows more operative flexibility and reduces the risk of extravasation, considering that the paste should be allowed to set prior to definitive hardware placement. All the fractures healed in both groups. The authors reported significantly lower rates of mechanical failure in the augmented group compared to the non-augmented group (1/29, 4.8% vs. 6/21, 28.6%, *p* <0.05). They evaluated loss of reduction on plain radiographs as the decrease of the distance from the top of the plate and the top of the humeral head. In the augmented group it was 1.5 ± 0.3 mm and was lower than that of the non-augmented group (2.59 ± 0.4 mm). This difference was statistically significant (*p* < 0.01). On the other hand, at 12 months follow up, there was no difference (*p* > 0.05) between the groups according the clinical outcome. In both groups, patients reported more than 75% of good to excellent results in terms of Neer functional score. Another CaS injectable cement (Stimulan, Biocomposites, United Kingdom) was used by Somasundaram et al. to augment 22 PHFs treated with plating fixation. At minimum 1 year follow up, no mechanical complications were reported by the authors [39]. The mean Disabilities of the Arm, Shoulder and Hand (DASH) score was 16.18, and the mean constant score was 64 points. In all patients, the calcium sulfate bone substitute was resorbed and replaced by trabecular bone after a mean period of 6 months.

Calcium sulfate is brittle, and injectable CaS cements have low compressive strengths that are more similar to those of cancellous bone than cortical bone [48]. Therefore, its use is limited to fill bone voids and should be avoided when structural support is needed, such as in medial cortex deficiency fractures. Moreover, calcium sulfates degrade rapidly and independently from bone formation [65]. Due to this rapid degradation, there is a risk that the loss of strength will occur too rapidly to properly support bone union and consequently lead to failure. Ideal bone substitute should have no stimulation of the surrounding tissues. CaS sulfate degradation may affect osteoconduction and cause an inflammatory reaction, including short-term drainage from the wound [66,67]. CaS degradation could locally release sulfur ions and alter pH [68]. More specifically, increased acidity related to dissolution of CaS has been postulated to cause localized demineralization [69]. Currently, none of these complications have been reported for the use of CaS injectable cements in PHFs.

### 3.2. Metallic Devices for Augmentation

Proximal humeral fracture augmentation could be achieved with the application of different mechanical devices that, despite different design, share similar biomechanical principles. In the literature, two systems have been used in the setting of PHFs: (1) The Da Vinci System or “triangular block bridge”, and (2) The Proximal Humerus Cage or “intramedullary cage” [29,42]. These implants aim to provide structural support to the humeral head and fill the metaphyseal void. Additional features (i.e., tuberosities fixation) are available for each of the two systems. Three retrospective studies described the use of the Da Vinci System for two-, three-, and four-part PHFs in a cohort of 78 patients, with a mean age of 51.4 years (range of 35–74 years) and mean 72 months follow-up (range of 12–132 months). One study retrospectively evaluated the use of the Proximal Humerus Cage two-, three-, and four-part PHFs in 11 patients, with a mean age of 65.4 years (range of 45–89) and mean 54 weeks follow up (range of 49–61 weeks).

#### 3.2.1. The Da Vinci System or Triangular Block Bridge

The Da Vinci system or triangular block bridge is a titanium triangle-shaped open prism, whose opposite faces are pierced and jointed with three pegs, one for each vertex (Figure 3). The triangle shape represents the evolution of the triangle-shaped bone block, which is originally handcrafted from an allograft such as iliac crest bone. The shape evolved from the beginning to the definitive version of the titanium cage (produced by Lima, San Daniele del Friuli, Italy, up to 2008 and then by Arthrex, Naples, FL, USA) [29]. Because five different sizes are available, the choice of the correct size and the exact position are therefore important. Each vertex has to fit into the head, the greater tuberosity, and the shaft, with the hypotenuse turned towards the metaphysis. This positioning ensures medial support, stopping the head from sliding down and ensuring adequate fracture support. It acts both as an expander and a metallic bridge on which it is possible to reconstruct all the fragments using minimal osteosynthesis with k-wires, cannulated screws, and trans-osseous sutures. The device aims to increase the stability in fracture with metaphyseal bone loss and to increase the proximal humeral re-vascularization and healing. That is due to stable effect obtained with the association of medial support and lateral fixation. 

The rationale for this technique is based on three points:
Fill the void left in the humeral head after a fracture;It is important to have an accurate anatomical reconstruction of the medial column;The cage creates a bridge between the head, the tuberosities, and the shaft. This allows the surgeon to perform a stable osteosynthesis.

Russo et al. retrospectively evaluated a total of 71 patients with two-, three-, and four-part proximal humerus fractures in three different case series. In 2013, they published the results of a retrospective evaluation of 69 proximal humeral fractures treated with the Da Vinci System between 2005 and 2010 [40]. In all cases, a deltopectoral approach was performed. The titanium cage was used to fill the void of the metaphysis. The fixation was performed with minimal osteosynthesis (cannulated screws or k-wires) or a low profile plate when the surgeon considered it necessary.

In almost all the cases, allograft was used to fill the cage. The choice of the associated osteosynthesis was made on the basis of the type of fracture. In all cases, minimum follow up was 2 years (range of 24–72 months). The fractures were radiographically proven to heal in 68 of 69 patients. There were two cases of malunion (one greater tuberosities dislocation, the other was a varus malposition of the head), and five patients suffered partial AVN. Because one patient suffered a deep infection, the device was removed and the patient was treated with a cemented antibiotic spacer. At the final follow up, the mean constant score was 80.25. Later, in 2017, the authors updated the series with a longer follow-up, comparing the results obtained with The Da Vinci System to the augmentation with autograft tricortical iliac crest or hand-shaped bone bank block [41]. In both groups, good clinical and radiographic results were reported, with a very low rate of complication and no statistically significant differences.

#### 3.2.2. Intramedullary Cage

Proximal Humerus Cage (Conventus Orthopaedics, Maple Grove, MN, USA) is an expendable nitinol wire cage designed to fill the void of bone in the humeral metaphysis [70]. Nitinol is a nickel-titanium, shape-memory alloy, characterized by an elastic modulus ranging from 75 to 83 GPa. The cage provides a stable platform to support the humeral head and the tuberosities, allowing nearly unlimited angles from which locking screws could be inserted to synthesize the bony fragments.

The cage can be inserted in an anterograde or retrograde direction and both percutaneously or with a traditional deltopectoral approach. The surgical technique involves the reduction of the humeral head that can be kept in place with k-wire, but it typically interferes with the deployment of the cage [71]. When a satisfactory temporary reduction is obtained, a 2.0 mm k-wire is placed from the distal lateral cortex into the humeral head. This can be done freehand or through a hole of a plate if supplementary plate fixation is considered necessary. The correct cage size is confirmed on the basis of the depth of insertion. The k-wire is then used as a guide for a cannulated 8.2 mm drill used to perforate the lateral cortex. On the k-wire, an expandable reaming device is then placed, which is used to ream the proximal humerus. The reaming device is then removed, and the cage is inserted and expanded to the previously reamed dimension. Nevertheless the reaming procedure provides additional augmentation through auto-grafting, which is entrapped inside the cage. The screws used for the lateral fixation do not have the role of supporting the humeral head because this function is performed by the cage itself. Therefore, screws need only to be long enough to pass through the cage in order to become a fixed-angle device, and the risk of screws cut out is reduced.

Hudges et al. in 2019 published the first study about this device, a series of 11 patients with three- and four-part PHFs [42]. The mean age was 58.5 years (range of 43–68), and mean follow up was 54 weeks (range of 49–61 weeks). Six patients received the cage alone (or with cannulated screws for the tuberosities), and five received the cage plus plate fixation. Three postoperative complications occurred, one caused by a fall of the patient in the immediate post-operative time, and the second was a case of avascular necrosis that subsequently was converted to a reverse shoulder arthroplasty. In the last case, the patient had hand swelling and persistent pain treated with stellar ganglion block. At 1 year follow up, patients reported excellent clinical results (mean Subjective Shoulder Value score of 69 and mean American Shoulder and Elbow Surgeons (ASES) score of 80).

### 3.3. Future Directions

Tissue engineering represents a modern alternative approach to current conventional treatment, which has found wide application for bone healing enhancement in orthopedics and trauma surgery [72,73]. Bone tissue engineering combines cells with regenerative potential (i.e., mesenchymal stem cells, MSC; endothelial progenitor cells, EPC), synthetic (i.e., β-tricalciumphosphate, β-TCP), or natural (xenograft, allograft, autograft) scaffolds and growth factors (bone morphogenetic protein (BMP) 2, BMP 7, vascular endothelial growth factor) [72,74]. The first attempt to apply these principles to proximal humeral fractures was made by Seebach et al. in a phase-I clinical trial, who used β-TCP granules as a scaffold for autologous bone marrow-derived cells (BMC) [75]. The 10 patients included in their cohort had two-, three-, or four-fragment fractures and a mean age of 69.1 years (range of 64–76). BMCs were collected the day before the scheduled surgery, from 50 mL of bone marrow from the iliac crest and additional 27 mL of peripheral blood. The samples were then centrifugated, concentrated, washed, and filtrated, obtaining a final 12 mL BMC suspension. Then BMC suspension was injected into the bone defect with the β-TCP granules (granule size of 1.4–2.8 mm; Chronos, Synthes, West Chester, PA, USA) and the fracture was then fixed with plate osteosynthesis. Safety and feasibility of the procedure was investigated, and the authors did not report any side effects of bone marrow aspiration or local/systemic adverse reaction and complication. All the fractures healed within 12 weeks without any mechanical complication. Clinical evaluation after 12 weeks showed satisfying clinical results, even though longer clinical follow up, as well as radiological follow up, is needed to assess the efficacy of this methodology. On the basis of the promising results, the authors in 2016 started the recruitment for a phase II clinical trial (ClinicalTrials.gov identifier: NCT02803177), but final results have not yet been published.

## 4. Discussion

Over the years, different augmentation techniques have been proposed to obtain the best results for PHFs. All of these techniques have the same purpose in common—enhance the stability of the medial hinge, and give support to the humeral head in order to avoid varus collapse and subsequent complications such as avascular necrosis and screw migration. Auto- and allo-grafting represent the most used strategies for PHF augmentation and patient treatment. However, even if good to excellent outcomes have been reported in the literature, these options still lack long-term follow up and comparative studies [76]. Nevertheless, the main problem of its extensive use is the concerns about possible side donor morbidity of autografts, their availability, and the high costs of allografts.

Therefore, in the last decades, bioengineering efforts have tried to identify the biomaterials that could properly avoid the need for bone tissue grafts. PMMA cements had wide application in orthopedic and in trauma surgery to improve implant fixation in low quality bone. PMMA use for PHFs have spread recently, particularly in improving the screw-tip fixation, with good biomechanical and clinical results. The results reported in this review did not always show clear advantages when compared to non-augmented fixation techniques. The rate of mechanical complications, such as screw penetration, plate loosening, or loss of fracture reduction was generally lower for patients who received PMMA augmentation (ranging from 0% to 16%) [30,33]. However, the only RCT in the literature that was performed by Hengg et al. did not report any statistically significant difference between the PMMA-augmented plate and the non-augmented plate fixation, even though both groups achieved excellent clinical results [34]. The use of PMMA as a metaphyseal void filler for PHF is not described, because PMMA cement is not resorbed and does not have osteoconductive properties.

On the other hand, the calcium phosphate cements are versatile bone substitutes that have both structural and osteoconductive functions. They have been used for PHFs to improve both structural support of the plate fixation and to fill bone void in the metaphyseal region [35,36]. Compared to PMMA cements, the main advantages of CaP cements are the low curing temperature reached during setting and subsequent lower risk of bone and cartilage necrosis. In the only reported comparative retrospective study, plate fixation with CaP augmentation showed a significantly lower mechanical failure rate than non-augmented plates and cancellous graft augmented plates [36]. The use of calcium sulfate injectable composites provided consistent results in favor of their use for augmentation of PHFs treated with plate fixation. CaS has osteoconductive activity—it is resorbed and slowly replaced by cancellous bone. The overall mechanical complication rate among the patients who received CaS augmentation ranged from 4.8% to 7.1%, whereas in non-augmented PHFs, mechanical failure rate ranged from 12.9% to 28.6% [37,38]. Two main differences exist between these two CaP and CaS cements. On one hand, calcium sulfate has a lower compressive strength than calcium phosphate cements. On the other hand, it is completely resorbed in a varying period of 4 weeks to 6 months, whereas CaP cement resorption could last for 10 years without sign of new bone formation. Therefore, CaP cements could lack osteoconduction and new bone ingrowth, whereas CaS rapid resorption could lead to lack of mechanical support, local PH alteration, and tissue inflammatory reaction.

In the literature, two different metallic devices have been used for PHF augmentation. The Da Vinci System is a titanium alloy prism block, which provides additional structural support for medial and lateral cortex. It is mainly used for the humeral head and for the two tuberosities. The results of the application of this device originate from a unique cohort of patients with good clinical results and low complication rates when compared to an allograft augmentation technique [40,41]. The Proximal Humerus Cage is a nitinol expandable intramedullary cage, which similarly acts as an internal support to the humeral head [42]. Here, additional autografting is provided by reaming of the medullary canal prior to the insertion of the cage. Results of these device are limited to a single case series of 11 patients. Therefore, even if the authors reported good clinical results and no complications, further studies are needed to prove the safety and feasibility of the techniques.

## 5. Conclusions

The available data support the effectiveness of PMMA, calcium phosphate, and calcium sulfate cement augmentation, which seems to be reproducible and safe when associated with conventional fixation techniques. PMMA showed the worst performances in terms of biological response because itis not resorbed by the host bone and has a number of potential issues regarding the risk of bone necrosis and leakage in soft tissues. Lower mechanical failure rates were reported after calcium phosphate and calcium sulfate augmentation. Calcium phosphate cements have the main advantage of being fully resorbed and replaced by new cancellous bone. Metallic cages have been used as mechanical metaphyseal support for PHFs only in a few case series, showing good preliminary results. However the level of evidence produced by the literature is not sufficient to recommend the extensive use of these procedures for PHF augmentation. Further studies are necessary to prove their efficacy compared to other existing techniques, in which cases represent a real advantage over the more common fixation technique. Future perspectives should take into consideration combined bioengineering strategies such as cell therapies and synthetic scaffolds.

## Figures and Tables

**Figure 1 jfb-11-00029-f001:**
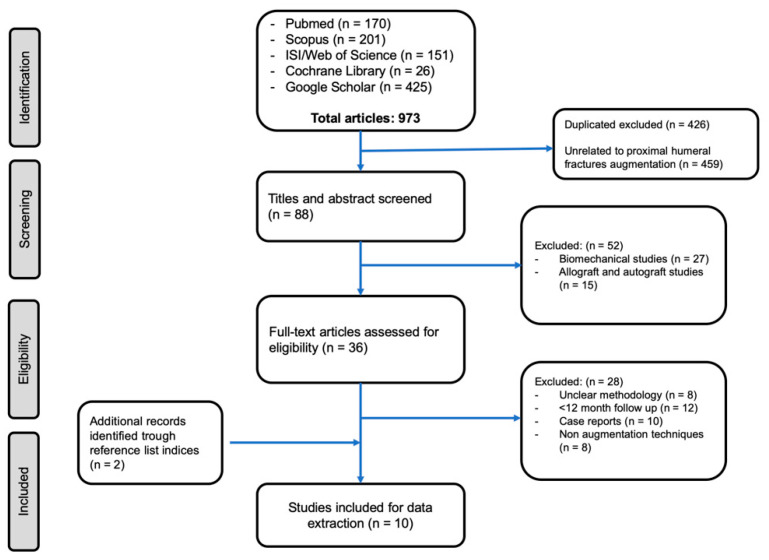
Preferred Reporting Items for Systematic Reviews and Meta-Analyses (PRISMA) flow diagram of studies’ screening and selection.

**Figure 2 jfb-11-00029-f002:**
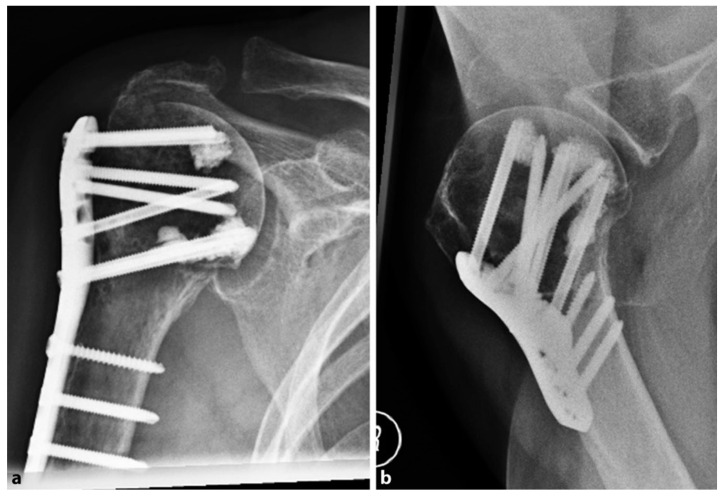
Postoperative radiographs of an 87-year-old female patient after locked plating of a proximal humeral fracture with additional cement augmentation of the anterosuperior and inferior humeral head screws: (**a**) anteroposterior view; (**b**) axillary view. Credit: Figure 3 from Kattaghen et al. [30] http://creativecommons.org/licenses/by/4.0/.

**Figure 3 jfb-11-00029-f003:**
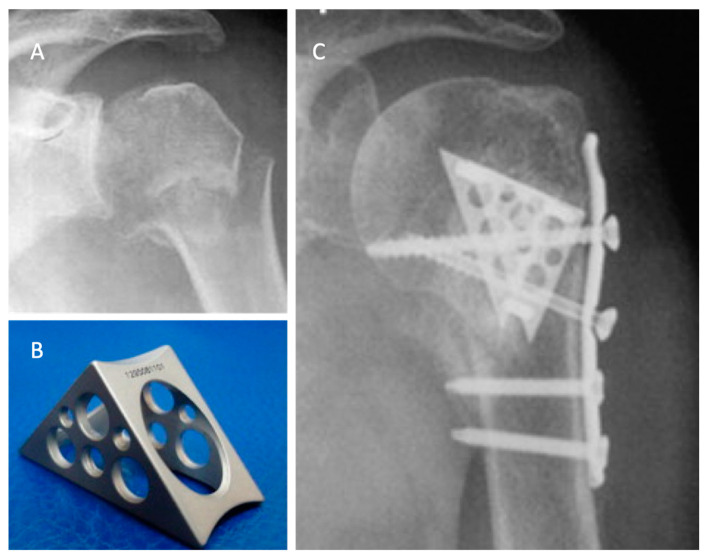
The use of a metallic cage for a proximal humeral fracture associated to plate fixation: (**A**) pre-operative X-ray of a displaced four-part humeral fracture-dislocation in a right shoulder; (**B**) The Da Vinci cage; (**C**) post-operative X-ray. Credit: Figure 3 from Russo et al. [40], permission conveyed through Copyright Clearance Center, Inc. (Order license ID 1027717-1).

**Table 1 jfb-11-00029-t001:** The articles included in the systematic review, listed by topic, level of evidence (LoE), and Coleman Methodology Score (CMS).

Augmentation Technique	Author	Study Type	LoE	CMS
*Synthetic bone substitutes and cement augmentation*				
Polymethyl methacrylate (PMMA) cement	Katthagen et al. [30]	Prospective non-randomized non-controlled	II	37
	Siebenburger et al. [33]	Randomized controlled trial	I	68
	Hengg et al. [34]	Retrospective comparative	III	52
Calcium phosphate				
	Robinson et al. (2003) [35]	Retrospective	IV	46
	Egol et al. (2012) [36]	Retrospective comparative	III	50
Calcium sulfate				
	Lee et al. (2009) [37]	Retrospective comparative	III	45
	Liu et al. (2011) [38]	Retrospective comparative	III	48
	Somasundaram et al. (2013) [39]	Retrospective	IV	33
*Mechanical devices*				
Triangular block bridge	Russo et al. (2013, 2017) [40,41]	Retrospective comparative	III	39
Intramedullary cage	Hudgens et al. (2019) [42]	Retrospective	IV	30

**Table 2 jfb-11-00029-t002:** Characteristics of the injectable biomaterials used for proximal humeral fracture augmentation in clinical setting [43,44,45,46].

Synthetic Bone Substitute Cements	Setting Time at Body °C	Compressive Strength	Young’s Modulus	Density	Porosity	Resorption Time	Osteoconductivity
Polymethyl methacrylate (PMMA) Traumacem V ^a^	9–15 min	85–110 MPa	1.9–3.0 GPa	1.18 mg/mm^3^	3%–5.4%	n/a	none
Calcium phosphate Norian SRS ^b^ Hydroset ^c^	4.5–10 min	36–66 MPa	674–790 MPa	1.29–1.78 mg/mm^3^	0.4%–0.6%	6 month to 10 years	low
Calcium sulfate MIIG 115 ^d^ MIIG X3 ^e^ Stimulan ^f^	5–11 min	10–40 MPa	665 MPa	≈2 mg/mm^3^	≈0.50%	6 weeks to 3 months	moderate

^a^ Trauma Cem V.; DePuySynthes, West Chester, PA, USA; ^b^ Norian Skeletal Repair System, SRS Cupertino, CA, USA; ^c^ Hydroset; Stryker, Mahwah, NJ, USA; ^d^ MIIG 115; Wright Medical Technology, Arlington, USA; ^e^ MIIG X3; Wright Medical Technology, Arlington, USA; ^f^ Stimulan, Biocomposites, United Kingdom.

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
