# Peer review of "Synthetic Bone Substitutes and Mechanical Devices for the Augmentation of Osteoporotic Proximal Humeral Fractures: A Systematic Review of Clinical Studies"

_jfb, 2020, doi:10.3390/jfb11020029_

Round 1
Reviewer 1 Report
Thank you for submitting this manuscript.
This manuscript is well-written in plain English, reviewed and summarized synthetic bone substitutes and mechanical devices for the augmentation of osteoporotic proximal humeral fractures.
However, there are some points to ask to the authors.
- This systematic review selected 10 clinical studies according to PRISMA guideline, 1 RCT and 1 non-RCT, and 8 retrospective studies. Is there a reason why there were scarce RCTs in this topic.
- PMMA has been used in many fields to improve screw fixation and fill voids, and provide structural support. However, there were no significant difference between fixation with and without PMMA as you described in three studies. Is there any benefit using PMMA which lack any usefullnes but several limitations?
- Some schematic or summarizing figures would be helpful to understand, especially for the metallic device for augmentation.
- Miswording on line 100, "LoE 3: 10 studies"
Author Response
Response to reviewer n.1
This manuscript is well-written in plain English, reviewed and summarized synthetic bone substitutes and mechanical devices for the augmentation of osteoporotic proximal humeral fractures.
However, there are some points to ask to the authors.
Dear Reviewer, we really appreciated your comments and we provided full response to all of your questions.
- This systematic review selected 10 clinical studies according to PRISMA guideline, 1 RCT and 1 non-RCT, and 8 retrospective studies. Is there a reason why there were scarce RCTs in this topic.
- PMMA has been used in many fields to improve screw fixation and fill voids, and provide structural support. However, there were no significant difference between fixation with and without PMMA as you described in three studies. Is there any benefit using PMMA which lack any usefullnes but several limitations?
I agree with you that there is lack of papers regarding this topic. Literature has been focused mainly on other skeletal districts (i.e. proximal femur and proximal tibia) in which fragility (osteoporotic) fractures occur frequently. In these fractures the benefits have been clearly reported.
The unsolved problem of humeral fractures outcome pushed researchers and the bioengineering industry to use bone substitutes (and lately mechanical device) for humeral fractures too in order to reduce the high failure rates of surgical treatment. Therefore it will take time since these therapies have wider application and will become the gold standard in humeral fractures too.
Our intention was to highlight the role of not-biologic augmentation in humeral fractures. At the moment allografts are considered the gold standard. However, many institutions (ours too) have difficulties in providing bone grafts. Therefore synthetic and metallic devices could be useful tools.
PMMAs showed the “worst” performances in terms of biological response, and have a number of potential issues. Nevertheless the evidence produced in the literature did not report higher adverse events. In conclusion, the only potential benefit should be considered the augmented screw tip fixation, but results suggest that Calcium Sulphate and Calcium Phosphate cements could be more reliable and bone friendly.
We stated this concept at the end of the manuscript in the conclusion section (line 408 - 401)
- Some schematic or summarizing figures would be helpful to understand, especially for the metallic device for augmentation.
In order to ease the comprehension, particularly for the readers not from the orthopaedic field, we added the figure 1 and figure 2 describing cement augmentation and metallic devices.
- Miswording on line 100, "LoE 3: 10 studies"
We have resolved this issue
Reviewer 2 Report
MAJORS:
The authors do not provide an explicit statement of questions being addressed with reference to participants, interventions, comparisons, outcomes, and study design (PICOS). This means that PRISMA standards have not been followed. A combination of terms is not the method that should be used. It should be specific keywords. This search is not reproducible and the authors might have forgotten some papers.
The authors use the Coleman Methodology Score (CMS) but they do not use these data to discuss their results. For instance, most of the studies are under 50 and are considered poor. Thus, they should be rejected. This lead to a total of 3 studies. Thus, this should be considered as a "Critically Appraised Topic" which downgrade the strength of the findings. To summarize, the conclusion is not supported by the evidence of the results.
OTHER COMMENTS:
They do not mention the dates of coverage and if they contacted authors to identify additional studies.
They should present a full electronic search strategy including any limits used, such that it could be repeated. This is not the case and the search cannot be repeated. Google Scholar is usually not considered as one of the databases that should be used. The authors should have used ISI Web ok Knowledge, SCOPUS, COCHRANE.
The authors did not mention who reviewed what, who is the expert authors in case of conflict.
The authors did not use Grey literature to find additional articles.
Author Response
Response to reviewer 2
Thank you very much for your deep revision of our manuscript. We evaluated with attention all the points you made and I think it led us to a great improvement of the overall quality of our work. I hope that our point by point response will be exhaustive.
I would like to specify that our intent was to provide an overview of the available data around this topic. Unfortunately, as often happens in the traumatological field, only few trials have been conducted. Therefore, due to the lack of evidence about clinical and radiological outcome, we choose to give to the reader (more likely a surgeon) additional information about the characteristics of the biomaterials used in this setting.
- The authors do not provide an explicit statement of questions being addressed with reference to participants, interventions, comparisons, outcomes, and study design (PICOS). This means that PRISMA standards have not been followed.
We totally understand your statement. In order to avoid redundancies we did not mention the PICOS. Now you will find at line 77 - 81.
- A combination of terms is not the method that should be used. It should be specific keywords. This search is not reproducible and the authors might have forgotten some papers.
Our search was performed using keywords (actually MeSH terms). See material and methods.
The authors use the Coleman Methodology Score (CMS) but they do not use these data to discuss their results. For instance, most of the studies are under 50 and are considered poor. Thus, they should be rejected. This lead to a total of 3 studies. Thus, this should be considered as a "Critically Appraised Topic" which downgrade the strength of the findings. To summarize, the conclusion is not supported by the evidence of the results.
I get your point. However, I don’t’ think that our conclusion tend to give a strong opinion in support of one or other technique. On the contrary, we have clarified from the beginning that there’s lack of evidence, that the overall quality of the studies is low and that, “Due to the substantial study heterogeneity and small sample sizes, the data obtained from the selected studies were not adequate to perform a meta-analysis. For these reasons, a descriptive approach to data analysis was performed”.
OTHER COMMENTS:
They do not mention the dates of coverage and if they contacted authors to identify additional studies.
We now have reported from first result to the last date of search (1966 – 31.12.2019)
They should present a full electronic search strategy including any limits used, such that it could be repeated. This is not the case and the search cannot be repeated. Google Scholar is usually not considered as one of the databases that should be used. The authors should have used ISI Web of Knowledge, SCOPUS, COCHRANE.
Our search was performed using keywords (actually MeSH terms). We performed an additional search on ISI and COCHRANE (with the same coverage period) that did not found any new paper. This because Google Scholar actually covers a lot of sources not listed in the other databases. Conversely many other remarkable authors reported the use of Google Scholar in well conducted Systematic Reviews.
The authors did not mention who reviewed what, who is the expert authors in case of conflict.
3 authors were assigned to this job. At least 2 investigators evaluated each article. If there was disagreement between reviewers, a third also reviewed the paper and the majority rating was used after discussion among reviewers.
The authors did not use Grey literature to find additional articles
I confirm that we did not use grey literature.
Reviewer 3 Report
I really enjoyed the writing of this article. In particular, it focuses on significant discoveries and gaps to be filled in order to bring this research to real applications, highlighting key points for future experimentation. The article is clear, the issue has been set and convincingly answered, adequately exploiting the data in the literature.
Author Response
Response to reviewer 3
I really enjoyed the writing of this article. In particular, it focuses on significant discoveries and gaps to be filled in order to bring this research to real applications, highlighting key points for future experimentation. The article is clear, the issue has been set and convincingly answered, adequately exploiting the data in the literature.
Thank you very much for this motivating good review. We tried to improve the paper adding two figures that describe cements and one of the metallic devices. Moreover, further correction were made according to the other reviewer suggestions.

Reviewer 4 Report
The content is good but lots of grammatical mistakes, cant be published in current form. Please go over the entire text in details. Eliminate the long sentences that have more than two commas and divide in 2-3 smaller sentences. Don’t place commas before ‘’and’. Don’t place transitional words (however, therefore and etc) in a middle of sentence, start new sentence instead. Reference number always at the end of a sentence. Avoid repetitions, e.g. in some sentences you have ``with`` 3 times.
Those are basic rules in the English scientific writing.

Author Response
Response to reviewer 4
The content is good but lots of grammatical mistakes, cant be published in current form. Please go over the entire text in details. Eliminate the long sentences that have more than two commas and divide in 2-3 smaller sentences. Don’t place commas before ‘’and’. Don’t place transitional words (however, therefore and etc) in a middle of sentence, start new sentence instead. Reference number always at the end of a sentence. Avoid repetitions, e.g. in some sentences you have ``with`` 3 times.
Thank you very much for this really useful review and the hard work you made to suggest these huge improvements to our paper. We have deeply rewritten the sentences according to your suggestions. I hope that you will enjoy this new version.
We have followed point by point your suggestion. Here you can find the detailed comments.
Those are basic rules in the English scientific writing.
- however. Is grammatically wrong, please correct: corrected
28, 29. good biocompatibility, osteoconductivity and lower mechanical failure rates when compared to non-augmented fractures: corrected
- Please check the Keywords for correct MeSH terms: corrected
- ,but according the literature,: corrected
- screws, : corrected
60-64. This applies for inhouse bone banks. The commercially available allografts may be more expensive, but they undergo strict control for eliminating the causes for infection, disease transmission and immunological reactions. Please add more literature and explainthis difference.: we added new references and explained the process of graft preparation
- remove , before and: done
- remove , as well.: done
- delete ‘’which are mainly synthetic bone substitutes’’ because there is no ideal biomaterial and you also explain that in line 117.: done
- differ from each other in mechanical and….:corrected
- remove , before and: done
129-130. augmentation [ref]. These studies had a total of….
- plate,
- rewrite, you cant have 2x ‘’with’ in 1 sentence with only 2 words apart….in 39 patients:I have rewritten the sentence
- mean….what?: I have rewritten the sentence
148-149. ;p>0.05)????: I have rewritten the sentence
149-151. divide in 2 sentences: done
- remove , before and: done
- remove , before the numbers and close with ). Start new sentence: done
- any explanation why was terminated?:The study was terminated due to the preliminary results of the trial. PMMA augmentation was not associated with lower complications rate or better function than conventional plating.
170-171. ,doesn’t have osteoinductive properties, also it is not integrated and reabsorbed by the bone. Therefore, this could….
163-172. Please add more references at such critical paragraph, especially on the issues.
169-170. Can you speculate why adverse events were not reported?The only reasonable speculation is that these are not common complication in the proximal humerus due to the large metaphyseal space and little amount of cement used for the augmentation procedure. However, this represents my idea (and my personal surgical experience) so I wouldn’t report it in the paper.
- Calcium phosphate cements: done
- and have lower: done
179-180. CaP cements can be reabsorbed and replaced by cancellous bone within 6 months to 10 years, which is why they lack predictable osteoconductive properties.: corrected
- why new paragraph after 1 sentence only?: corrected
184.185. having severely ……by internal fixation: corrected
- composite. It is applied in a: done
- range 5 to 10:done
- maintained and: done
195-198. rewrite the entire sentence like: Egol at al in 2012 used the same setting with another….: done
- Twenty seven…..never start a sentence with number: done
- calcium sulphate group?: corrected
- In comparison, significantly higher rates …were found: done
- Only 1 complication occurred in …..due to deep infection: done
- what do you mean by authors didn’t report any clinical result?: I mean that the authors did not report any functional score, ROM analysis etc. I change the sentence at line 222-223. “The authors did not reported the functional outcome of the patients using clinical scores, pain scales or range of movement analysis”
- please end with paragraph for disadvantages just like you had for PMMA. For example, its degradable and has less volume stability if compared to PMMA. Degradation time of different brands…you said 6 months to 10 years….how is this good for reproducible results? Be creative, there is no ideal biomaterial!
Both phosphate and sulfate cements are biodegradable, why for phosphate you state not osteconductive, but sulfate it is. Yes, there is difference is degradation time but since are both replaced by bone then both are osteconductive.
See the integration line 224 – 235
- 3.1.3- NOT 4. Calcium sulphate cements: done
- successfully for treatment….we know its synthetic, its in the title: ok
- what is biointegration? osseointegration and osteoconductivity, but its….
- products such as faster…: ok done
- reabsorbed and replaced withing 6-12 weeks, which is why they are considered to have good osteoconductive properties. (not activity).: corrected
- For this review, we considered two eligible retrospective….series. Here total of 65 patients received …..: changed
- Here 14 out of 45….: done
- USA). The aim was to…..: done
- ), it requires a setting time….: done
- [35]…place at the end of sentence.: done
- Here the MIIG115….: done
232-236. Very long, split into 2-3 smaller sentences like: minutes). That allows more operative flexibility ….reduces …..: corrected
- there was no difference:
- put ref number at the end of sentence
- please end with paragraph for disadvantages….like its degradation leads to free sulfur that reacts with water and makes acidic environment-not very good for new bone formation….be a bit more critical, because these is no such case with CaP biomaterials, free Phosphorus is used for new bone formation, it macro-element.
3
Sulphur is in much smaller concentrations in the body and too much free sulfur at one location => local acidic environment.Please see line 276 - 286
- that despite…: done
- 35-74)….where is ( before that?: corrected
- allograft such as iliac crest bone.: done
- from the beginning to the: done
- Since 5 different…: done
- all the fragments: done
- revascularization and healing. That is due: done
- fixation.: done
- 2010 [37]. In all cases, a …..with minimum 2 years….: done
- The fractures were radiographically proven to heal in…: done
- ) and 5 patients….Since 1 patient: done
- In both groups,: done
- differences. ‘’among the 2 groups’’ is repetition: done
- put [56] at the end of sentence: done
- placed
- removed, as the cage is then
- place [39] at the end of sentence
- that subsequently ….arthroplasty. In the last case, the
- synthetic (ß-TCP, ….) or natural (xenograft, allograft, autograft) scaffolds and growth factors (BMP 2, BMP 7, VEGF……)…add some references here like Trajkovski B et al, Materials 2018, 11(2), 215; https://doi.org/10.3390/ma11020215: done
- augmentation and patient’s treatment. However,: done
- literature, these:done
- Nevertheless,: done
- autografts, their availability and the: done
- when compared to: done
- literature that was: done
- The use of PMMA: done
- described, because …..and doesn’t have ostecon….: done
- On the other hand, the calcium…. : done
- to improve both structural: done
369 cements are the…..reached during setting and: done
- it is resorbed: done
- Moreover, CS….then 372. CP….373. again CS……please rearrange sentences and make clear end for CP, then continue with CS like…On the other hand CS…..:I ve rearranged this part
- differences exist between CP and CS: done
- On one hand, CS has….. On the other hand, it is …..: done
- continue with why is that and how is that better, please explain….both mechanical and chemical aspects: please see line 431 - 433
- cortex. It is mainly used for: done
- rates when compared to: done
- cage is a …..acts as an: done
- Here additional: done
- patients. Therefore,: done
- supports the: done
- Both CP and CS cements have the ……when compared to PMMA.: done
- The role of….this is empty sentence, please explain what do you mean, is it well known, well reported, you need some verb here: rearrange the phrase
- technique and: done
References should be corrected, many are missing details of journal volume and pages: checked and corrected

Round 2
Reviewer 2 Report
The authors revised some aspects of the paper. However, the Reviewer still has some majors comments with their article. The study is still not reproducible, and there is no quantitative comparison of the outcomes. The authors should consider re-writing this as a critically appraised topic. Indeed, their study is more a summary of a search and critical appraisal of the literature related to a focused clinical question than a systematic review.
Majors comments:
A systematic review should be reproducible. When I enter all the keywords that the authors list in their papers:
((((((((augmentation) AND (humeral fracture proximal)) AND (polymethylmethacrylate, PMMA)) AND (cement)) AND (bone substitutes)) AND (hydroxyapatite)) AND (calcium phosphates)) AND (calcium sulfate)) AND (Cell- and 87 Tissue-Based Therapy)
No results appear on PubMed. Please advise which keywords have been used, how and with AND or OR. So far, the study is not reproducible.
Where is the comparison written? Relative risk reduction (RRR) should be calculated for both primary and secondary outcomes. Then, they should have a separate graph for intervention with similar design and outcomes.
The functional scores appear only in "Lee and Shin in 2009".
What are the radiological results? Where are the objectives measurements?
It seems that complication is the most reported outcomes, so it should be the primary outcomes and include in the table.
"At least two investigators evaluated each article," it should have been the same person. Please provide their initials and the initials of the third one.
Minor comments:
" The last date of the literature search was 31 December 2019" And the authors state that they have added two databases since the last review. Please correct.
"and the majority rating was used after discussion among reviewers" I am not sure to understand. If there is a disagreement, the third investigator should be the one making the decision.
In the Prisma flowchart, please state how many papers per database.
"Clinical results in terms of functional scores, radiological results, and complication rates systems were compared to non-augmented patients."
Author Response
Response to reviewer n.2
The authors revised some aspects of the paper. However, the Reviewer still has some majors comments with their article. The study is still not reproducible, and there is no quantitative comparison of the outcomes. The authors should consider re-writing this as a critically appraised topic. Indeed, their study is more a summary of a search and critical appraisal of the literature related to a focused clinical question than a systematic review.
Majors comments:
A systematic review should be reproducible. When I enter all the keywords that the authors list in their papers: ((((((((augmentation) AND (humeral fracture proximal)) AND (polymethylmethacrylate, PMMA)) AND (cement)) AND (bone substitutes)) AND (hydroxyapatite)) AND (calcium phosphates)) AND (calcium sulfate)) AND (Cell- and 87 Tissue-Based Therapy). No results appear on PubMed. Please advise which keywords have been used, how and with AND or OR. So far, the study is not reproducible.
- We agree with your comment. The search strategy was not well explained at all. Therefore we reported it line 86 – 89. “(humeral fracture proximal) AND (Bone substitutes OR augmentation OR hydroxyapatite OR cement OR PMMA OR calcium sulfate OR calcium phosphate OR cell therapy OR tissue engineering)”
Where is the comparison written? Relative risk reduction (RRR) should be calculated for both primary and secondary outcomes. Then, they should have a separate graph for intervention with similar design and outcomes. The functional scores appear only in "Lee and Shin in 2009". What are the radiological results? Where are the objectives measurements? It seems that complication is the most reported outcomes, so it should be the primary outcomes and include in the table.
- We agree with you that there is not comparison among the different outcomes reported by the studies. However, the studies evaluated clinical results through different methodologies and different outcome measures that cannot be objectively compared. I’m not confident into calculating Relative risk reduction (RRR) with this small and heterogenous cohort of patients (only 1 trial).
However, I would add the clinical outcomes reported for all the studies, even if they have used different functional scores. You can find the changes in the text in red font.
- I’ve changed the outcome hierarchy in line 82 – 83: “The primary outcomes were mechanical failure and complications rates compared to non-augmented fractures. Clinical results in terms of functional scores were also reported”
- The authors did not report radiological measurements (i.e. varus diplacement), but only fracture union rates and other complications. Therefore I will specifically refer to these as primary outcomes.
"At least two investigators evaluated each article," it should have been the same person. Please provide their initials and the initials of the third one.
- We provided the needed information at line 95 – 97)
Minor comments:
" The last date of the literature search was 31 December 2019" And the authors state that they have added two databases since the last review. Please correct.
- All the databases searches were limited to 2019.
"and the majority rating was used after discussion among reviewers" I am not sure to understand. If there is a disagreement, the third investigator should be the one making the decision.
- We provided the needed information at line 95 – 97 (If there was disagreement between reviewers, any discrepancies were resolved by a third reviewer (M.V.))
In the Prisma flowchart, please state how many papers per database.
- We changed it according to your suggestions
